# Designing a Large-Scale Immersive Visit in Architecture, Engineering, and Construction

Rachid Belaroussi [1,*], Huiying Dai [2], Elena Díaz González [3] and Jorge Martín Gutiérrez [3]

1   COSYS-GRETTIA, University Gustave Eiffel, F-77447 Marne-la-Vallée, France
2   ESIEE Paris, University Gustave Eiffel, F-77454 Marne-la-Vallée, France; huiying.dai@edu.esiee.fr
3   Higher School of Engineering and Technology, Universidad de La Laguna,
    38071 San Cristóbal de La Laguna, Spain; elediaz@ull.edu.es (E.D.G.); jmargu@ull.edu.es (J.M.G.)
*   Correspondence: rachid.belaroussi@univ-eiffel.fr

**Abstract:** Throughout history, tools for engineering in the building industry have evolved. Due to the arrival of Industry 4.0, Computer-Aided Design (CAD) and Building Information Modeling (BIM) software have replaced the usage of pens, pencils, and paper in the design process. This paper describes the work required to design a large-scale immersive visit of a district under construction in a suburban area of Greater Paris, France. As part of this real estate project, called LaVallée, we have access to its city information model: all the BIMs of the works to be carried out including roads, terrain, street furniture, fountains, and landscaping. This paper describes all the technical operations necessary for the design of an immersive 3D model with a high level of detail of the neighborhood with its surroundings. The objective of this technical report was to provide practitioners with feedback on such an achievement based on industrial-level data. The development of the city model begins with the registration of all the BIMs from different firms in a common Geographic Information System: this gives the opportunity to confront the operational requirement of a construction phase and the actual current practice of architecture firms. A first prototype was developed using the archviz tool TwinMotion. In order to increase the realism of the model, we describe the creation of a pipeline in Unreal Engine with the automated tasks of material and mesh replacement and the lighting and landscape configuration. The main contribution of this work is to give relevant experience on building such a large-scale model, with the Python script when possible, as well as the necessary manual steps. It is a valuable contribution to the making of large-scale immersive visits with a high level of detail and their requirements.

**Keywords:** immersive visit; digital twin; building information model; city model; architecture; Unreal Engine

## 1. Introduction

### 1.1. Purpose of the Work: An Immersive Visit at the Scale of a Neighborhood

For the construction industry to be completely digital, structured information models would need to be available at the construction site, where the information is used to shape the material world. The design phase is mostly digital and is becoming more and more integrated with BIM.

The presented work in this paper is part of a larger research program of Eco-district Smart, Sure, Sustainable. The program uses the opportunity of a large construction project called *LaVallée* recently started in Paris's suburbs, France. The district's spatial extent is about 500 m × 400 m, part of a larger city called Châtenay-Malabry. The whole district is to be completed in 2025. We are in the first phase of the project with the first roads being built and the first inhabitants installed by the end of 2022. Eiffage, the company in charge of the real estate development, has cofunded a partnership with University Gustave Eiffel. This allows us to have access to all the construction details. In this work, we had access to all

the BIM files from which we intend to maintain a complete 3D model of the future district by integrating the new BIM files as soon as they are available. This paper describes the various operations required to build such a large-scale model, with a close-to-photorealistic appearance, and how we were able to automate the process.

Figure 1 illustrates the virtual future look of LaVallée district in its urban context: the purpose of the work is to build an immersive visit of the future district based on professional BIM architectural files of its buildings.

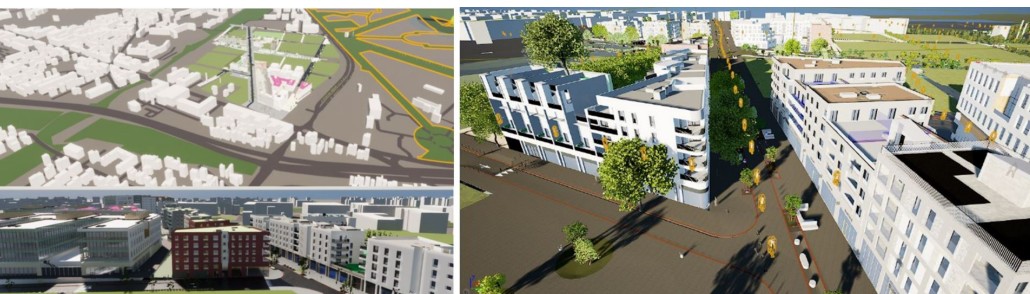

**Figure 1.** LaVallée district: location in the city and point of views of the large-scale immersive visit.

In its design, the LaVallée district is physically open to the outside and will offer services that will be of interest to other residents or users of the surrounding area. To know the effect of this opening on the potential transit of visitors in the district, as well as the places of interest for the inhabitants, it is necessary to design visualization tools of the district in a project situation (i.e., once the eco-district is built).

In this work about the future appearance of the neighborhood, some general objectives are listed below:

- Visualize the details of the future district in 3D in an immersive tour with virtual reality glasses (HTC Vive).
- Develop tools to automatically extract basic elements from the high-precision CIM of the project and integrate the predicted motion of people.
- Predict and characterize the quality of the built environment and urban ambiance, in static and dynamic scenarios, based on human sensations in virtual reality.

The overall objective of the project is to provide an immersive visit of LaVallée in which an architectural ambiance study such as the one described in [1] can be conducted in virtual reality. It requires collecting the BIM files of over 30 buildings, registering them in a common georeference frame, and automating the material replacement, lighting setup, and landscape. This paper provides valuable feedback on the holistic approach required to set up such a large-scale 3D model with the realistic appearances of textures and lighting.

*1.2. Applications in Architecture, Engineering, and Construction*

Extended reality is a significant modality for the Architecture, Engineering, and Construction (AEC) sectors, because the constructed world is inextricably linked to three-dimensional space and because AEC professionals depend significantly on imagery for communication.

Six broad use cases have been identified:

- Stakeholder engagement: Extended reality may be used to engage potential clients or the general public in order to create a more realistic version of a developed asset and to generate more relevant or informed feedback. It may also be quite beneficial in the real estate industry. The system will give professionals and their clients a complete picture of the project, reducing the risk of errors and unpleasant surprises. It will let potential clients inspect homes without physically being present.
- Design support: With the help of extended reality, designers may be able to see how their decisions will affect the final products and learn more about them.

- Design review: VR assist in the transmission of design intent, allowing for more efficient design review; problems can be spotted more easily, and sign-off can be completed more rapidly.
- Construction support: There are four areas where extended reality may be used to help with construction. The first is construction planning. In this field, the major goals of AR and VR are to predict future difficulties and enhance delivery. Virtual Reality (VR) focuses on generating immersive construction simulations, whereas Augmented Reality (AR) focuses on seeing virtual objects that can be built on-site. Tracking construction progress: This is vital because the early detection of schedule delays is essential for on-time delivery. In construction safety, VR can help to create safer working environments by assisting with danger detection and inspection. In the construction industry, when virtual reality is used for training, it reduces the risks that workers face and improves operations. The last area of involvement is operational assistance: VR provides tools on construction equipment teleoperation.
- Building operations and management: As AR may offer important information to site employees who run and maintain buildings, VR might allow for remote operation of the facility in an immersive environment. Both technologies may be used together to help field and remote office employees and increase cooperation.
- Training: Virtual reality may create realistic situations in which users gain knowledge and skills by simulating real-world actions. By mimicking the usage of expensive equipment, modeling risky locations, decreasing travel expenses, and increasing health and safety, both AR and VR technologies may minimize the cost of training.

Extended reality technologies, however, are not yet stable and dependable enough to meet real industrial requirements. One of the main factors limiting acceptance in the dynamic and severe environment of the construction industry is technical restrictions. One of them is when consumers have negative reactions to VR surroundings as a result of a misalignment between what they see and what they perceive. Another problem is cost. In fact, despite the fact that the cost of establishing VR systems has fallen, considerable financial investments are still required.

*1.3. Related Works: Large-Scale 3D Models in Urban Area*

In [2], a detailed review of the applications of 3D city models was given. This study showed that 3D city models are employed in about 30 use cases that are a part of a multitude of application domains for environmental simulations and decision support, valuable for several purposes beyond visualization. Interested readers can refer to [3] for digital twins from the point of view of the city stakeholders. We focused our work on 3D city models characterized by a high Level Of Detail (LOD), a measure that indicates their grade and scale, for architectural and art purposes mainly.

The 3D city models are derived from various acquisition techniques such as photogrammetry, radar and laser scanning [4–6], cadastral drawings [7], extrusion from 2D footprints [8], or crowdsourced geoinformation [9,10]; all are GIS-based sources of information that can sensed and measured because the infrastructure exists. Another more precise kind of data are architectural BIM models [11], which are high-level-of-detail models and are already available for future construction or real estate development in progress.

Few city models based on professional BIM are described in the literature. For instance, in [12], the authors linked an existing BIM to generic simulation models of construction activities implemented as script game components within a game engine, but only one BIM was used that modeled a sample house. A meta-analysis of the work available in the field of BIM's application can be found in [13]. However, as underlined in [14], there is limited literature about virtual reality technologies and BIM in AEC: the most-common uses of VR are safety training [15], project schedule control, collaborative projects [16], design issues, and construction site layout. However, to our knowledge, no publication has explained the necessary steps required for the building of a 3D city model based on professional BIM files.

Let us cite some examples of realizations with GIS data. Aerometrex created a 3D model of the entire city of Adelaide, using a combination of photogrammetry data captured via helicopters and the real-time archviz tool TwinMotion [17]. The firm used the real-time architectural visualization tool TwinMotion to create a 3D model of Adelaide and the area surrounding it—around 1000 sq km—using photogrammetry data captured via fixed-wing aircraft and helicopters.

AccuCities [18] designed a visualization of London's architectural evolution and the growth of the city over 1 km$^2$ tiles and its 3D model. The digital city model was captured from three separate aerial surveys conducted in 2016, 2019, and 2022. The 3D mapping operation created the 3D model using manual stereophotogrammetry. A simple 3D city model was first created by extruding building footprint polygons, then updated at a finer level of detail. They used the archviz tool Unreal Engine as a graphical engine.

The CityGML database 3DCityDB [19] is a repository of free 3D models useful for urban context display. These are low-level-of-detail models useful to integrate with your high-LOD architectural BIM. Interested users can see, for instance, a 3D city model of Berlin portaldeveloped by the Technische Universität München [20]. The interface exploits publicly available open data in CityGML format, which contain around 550,000 LoD2 building objects (basic roof shape and orientation) within the whole city area (890 km$^2$).

In [21], the authors described how to derive CityGML building models from BIM data. The conversion of BIM data to GIS data has several challenges due to the differences in the reference system, level of granularity, and geometric representation methods, as well as semantic mismatches between BIM and GIS data models. Our approach is the other way around, since we only integrated GIS data for the urban context of LaVallée into our city model made of a collection of BIM data.

BIM models are high-fidelity three-dimensional construction plans; they are seldomly used in large city models because of their relatively large size: typically, a simple four-story building's BIM can be 300 Mo large or much bigger. Therefore, practitioners usually use lower LOD representations, which are more than enough for urban planning purposes, but are not refined enough to study architectural ambiance from a human centric point of view, which requires a high level of finesse. This paper gives an example of what is required to build a 3D city model from BIM data.

## 2. Design Justification and Research Gap

### 2.1. From GIS-Based to Professional BIM-Based City Model

Most of the papers available in the literature about digital twins describe the construction of large-scale 3D models based on GIS data. These models are accurate at a mesoscopic level, the level of the city, which is adequate for urban planning purposes, but still, they are low-LOD models. These are typically 2.5D models consisting of a 2D footprint of a building with its elevation, as shown in Figure 2a, often times with a coarse texture extracted from aerial photogrammetry or from pictures acquired from a ground vehicle. Researchers expose their work designing their own tools or using common GIS software. The emphasis is more on the city scale and number of buildings integrated than on their aesthetics. If one looks at designing an immersive visit, a level of detail much greater is required, that is where BIM Level 4 is required. Figure 2b gives an example of the refinement that can be reached with BIM-based models.

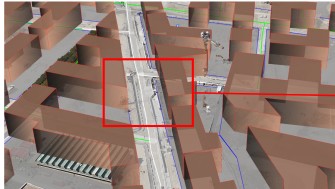 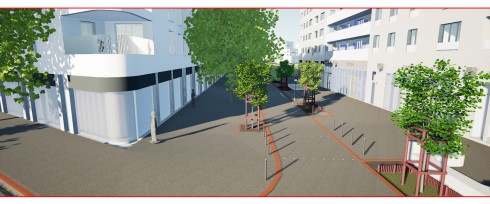

(a) Level of Detail LOD 1 – GIS data      (b) Level of Detail LOD 4 – BIM data

**Figure 2.** Level of detail of GIS- and BIM-based 3D models. (**a**) GIS city models are 2.5D models: footprint plus elevation, with textures. (**b**) An immersive visit requires a high level of detail.

While GIS-based approaches are well documented, there is a lack of information on how a practitioner would proceed if he/she wanted to build a 3D city model from professional BIM files. AEC consultants use commercial tools such as the cloud-based technology BIM 360 [22], which facilitates the visualization of the city and existing infrastructure. Figure 3a illustrates a point of view of our district of interest: in this parcel are represented artwork from several buildings, as well as urban furniture data and landscape data (basically trees). For architectural consideration, BIM 360 is a powerful tool from a functional point of view, but the textures are mainly representative of what one can find in a typical virtual world. Such a digital twin is well adopted for design review, construction support, and stakeholder engagement [23], but lacks realism. It is useful for interaction, but personal imagination is needed to fill in the gaps of the cold textures' impression and improve the immersive experience. Figure 3b illustrates the kind of results that are required for an immersive visit that would allow aesthetic and architectural ambiance study: much work is required in terms of texture replacement and lighting configuration. The game industry is able to deliver such a result using game engines such as Epic Games or Unity technologies [24], but game designers design graphically their own buildings without the need for BIM data, which is more a work of art than an integration of industrial processes.

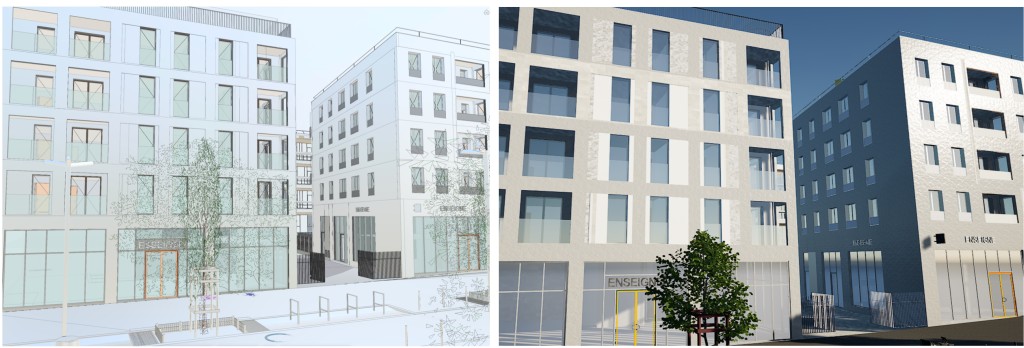

(a) BIM 360: virtual functional view　　　　(b) Unreal Engine: virtual photorealistic visit

**Figure 3.** (**a**) AEC traditional view of buildings: perfectly functional, but cold textured. (**b**) The same view produced with a game engine, enhanced in photorealism for a more immersive visit.

### 2.2. Need for Photorealism

Therefore, there is a lack of information on the process required to build a 3D city model from a collection of professional BIMs, especially in the case of a real-world scenario of a large-scale real estate development. For a district designed to house 6000 residents, the number of AEC firms involved can be very important and the BIM practices can vary from one company to the other: even with centralized BIM management, some heterogeneity persists in the production of BIM data.

Another gap to be filled is the visual realism of the digital twin produced from BIMs: most of the AEC market is focused on the interaction and functionality of the 3D models. The digital twin should represent all the functions such as facades, gates, accesses, staircases, urban furniture, and vegetation in a geometrical way fitting the final product, but with little regard for the realism of their appearance, as illustrated by Figure 3. While realism is correctly established for indoor environments, as in the case of the virtual tour of the inside of an apartment, it is problematic if one wants to build a truly immersive visit of the neighborhood, its outdoor environment, and its public places. Architects are invested in the design of attractive spaces, but must also consider the senses and emotions that users may feel through the shapes and material, with maximum expressiveness in a minimum space. To do so, they can hire a firm to produce a cinematic marketing model of their building, but this would still need their integration with the work of other companies in charge of adjacent buildings for a holistic human experience. The subject of architectural ambiance using virtual reality is seldom tackled in the literature, usually addressed with mock-ups of structures representing buildings, but never with professional production,

such as BIM files. The subjective feeling of presence is determined in a virtual environment by immersion and realism, which have to be integrated in a human experience in the perception of architectural spaces [25]. It is a necessary part for the study of the neuro-architecture concept on how architecture can affect our well-being: physically, intellectually, and emotionally [26]. However, to the best of our knowledge, no effort has been made in the literature to document the extension of a BIM-based model to a photorealistic architectural 3D model, especially in the case of a large-scale area requiring a large part of automation in this task.

### 2.3. Methodology

To examine BIM as a means of project integration in smart city development, a case study of a project utilizing Revit, BIM 360, TwinMotion, and Unreal Engine for the immersive visit of a future neighborhood currently under construction was performed. This project involved existing GIS buildings models in a suburban city of Greater Paris, France. A total of 30 individual Revit building models were used for the project. For this study, the digital twin development process was evaluated and documented. The scope of the smart city is the city of Châtenay-Malabry in the south of Paris's suburbs.

Figure 4 illustrates the workflow of the design: it goes from AEC software for handling architectural data to a game engine for photorealism and features' augmentation. It starts with the collection of the 3D models by the BIM manager from all the AEC partners of the project. Each company is responsible for the CAD drawings of the buildings, road networks, terrain, public places, parking spots, urban furniture including lighting and benches, park and recreation furniture, fountains, and vegetation. The data are centralized through the BIM 360 cloud platform. All BIMs are registered to a common coordinate system with Revit and exported to a TwinMotion project. With this archviz software, it is possible to augment the project with the urban context of the real estate project, as a form of GIS model of the surrounding buildings and the network infrastructure of the city. A first prototype was built, which is very useful for rapid checking and first-hand visualization including the possibility of using VR head-mounted device. Yet, this prototype has been deemed too colorful and not realistic enough for a perfect immersion experience. Therefore, we decided to use the game engine Unreal Engine for the final part of the immersive visit design. In this article, we describe the automation of the tasks through Python scripts to render a more photorealistic architecture model of the district. These tasks included the loading of the TwinMotion prototype from the disk, material replacement for the buildings and vegetation mesh replacement, as well as the lighting configuration of the complete scene.

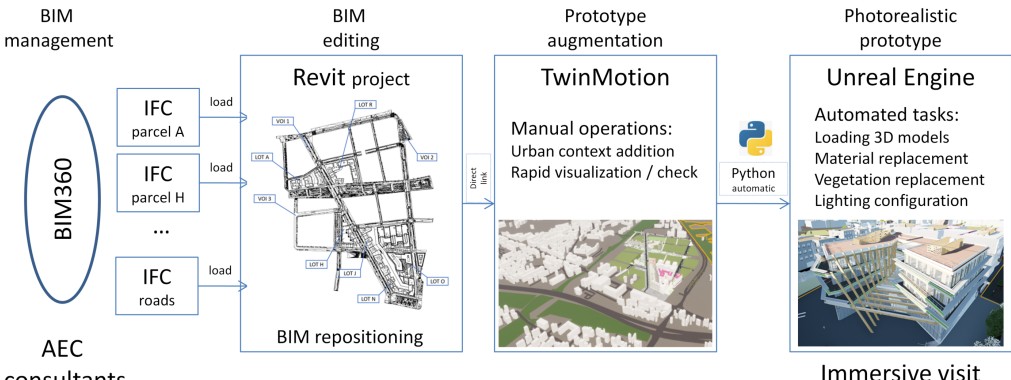

**Figure 4.** Immersive visit design: workflow diagram from AEC artwork data to virtual reality setup.

## 3. Context: Real Estate Development and Urban Living Lab

### 3.1. LaVallée: An Urban District under Construction

LaVallée is a real estate development project under construction. The district is located in a southern suburb of Paris France, specifically in Châtenay-Malabry city, and its spatial extent is approximately 500 m × 400 m. Figure 5 shows the blueprint of the

future neighborhood displayed over the current construction site. The construction work in the area began in 2018 and is expected to be completed by 2025. The real estate project aims to create an eco-housing district, with a childcare center, school system, secondary school, gymnasium, stores, an urban farm, and other facilities, which will include more than 2000 housing units. All of these facilities, which are accompanied by manicured public areas, embody the essence of the project: a new neighborhood that is concerned with the quality of life of its residents.

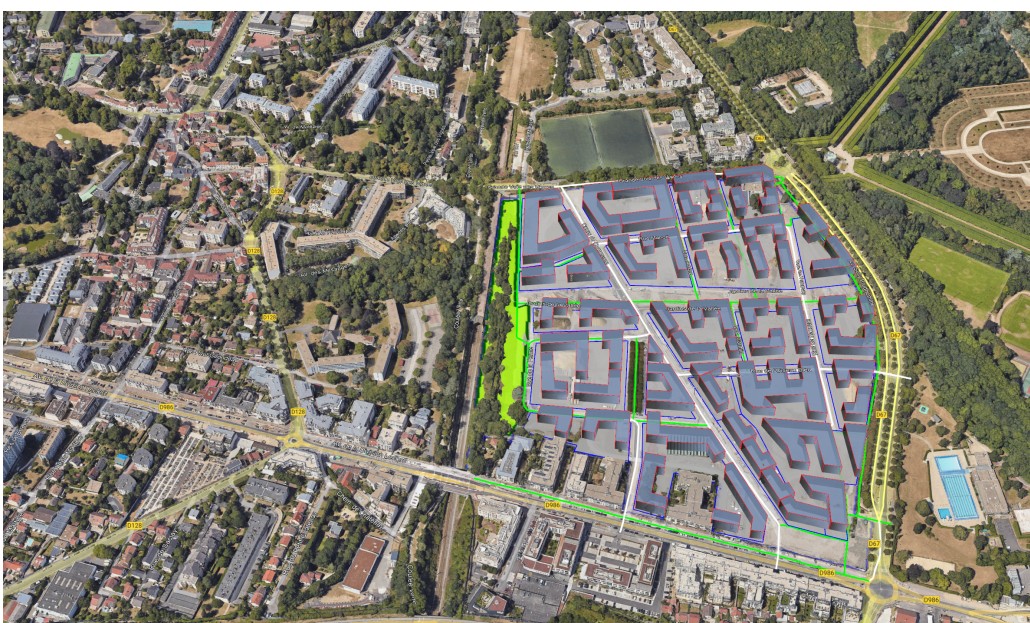

**Figure 5.** Mass plan of the future LaVallée neighborhood and its situation in the city. Aerial image: Google Earth Pro.

This enables the city to strengthen its urban and green identity, to revitalize a neglected space at the city's entrance, to ensure urban connections with adjacent neighborhoods and the Parc de Sceaux, and to provide an architectural signature by enhancing the city's qualities and heritage.

The program has about 213,000 square meters of surface with:

- 120,000 square meters for residential buildings;
- 24,000 square meters for social housing;
- 40,000 square meters for offices;
- 15,000 square meters for commercial space;
- 14,000 square meters for public utilities (college, school group, gym, nursery, etc.).

The land was purchased at the end of 2017 by the SEMOP group, a new model of partnership between public and private bodies. The public part is the municipality of Châtenay-Malabry, while the private part is the Eiffage company. Eiffage is the third-leading French group in the civil engineering and public works sectors. The SEMOP or Mixed Economy Company with Unique Operation is driven by an innovative legal structure: the first of its kind in France to be exclusively specialized in urban development. The shareholding of SEMOP is made up of Eiffage Aménagement (50%), the EPT Vallée Sud Grand Paris (34%), and the Caisse des Dépôts et Consignations (16%). Eiffage company is consequently responsible for the operation's finance and execution. The City, on its part, maintains political authority over the project, with the mayor chairing the SEMOP Supervisory Board and approving operational decisions. SEMOP teams are primarily responsible for the site's organization and coordination.

This large-scale project, which is being carried out by a number of different companies, has been divided into three phases of development (Figure 6), with the last phase due by 2025. As of 2022, the first phase is nearly complete, and the second phase will begin in 2022.

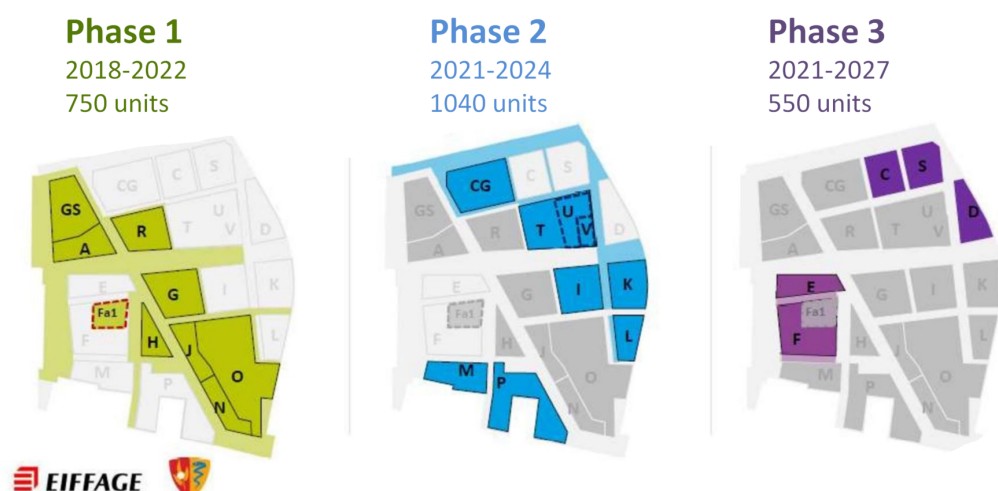

**Figure 6.** Three phases of development. ©SEMOP Châtenay-Malabry Parc Centrale.

The first phase of the project involves the full area surrounding the district's main street, Cours des Commerces, whilst Phases 2 and 3, by using the main street as the district's core, expand radially to the latter. Therefore, once Phase 1 is completed and the procedures for the second phase are followed, it is possible to bring in the first residents.

### 3.2. AEC Partners

Between the diverse landscapes of the existing districts in Châtenay-Malabry and the nearby public park, ensuring consistency is a challenge. The magnitude of the operation adds additional complexity: it will be constructed in several steps, and there will be numerous companies involved, all with different working cultures and in different fields: Architecture, Engineering, and Construction (AEC).

Aware of this pressing need for consistency, the SEMOP chose Arcadis, a global design, engineering, and management consulting company, to coordinate the project's various stakeholders using innovative City Information Management (CIM) processes.

These processes make it possible to create a digital mock-up of the project and share it with the partners. The collaborative mock-up thus allows the design to be followed and each building or public space in the project to be constructed. It comprises the entire project and its surroundings, spanning from the various neighboring districts of Châtenay to Sceaux Park. The mock-up thus allows Arcadis to ensure the overall harmony of the project as it progresses and the consistency of future buildings, not only with their environment, but also with each other. Moreover, it integrates all the data needed to manage flows (water, electricity, gas, etc.) and calculates their environmental impact.

Arcadis has not only implemented the collaborative digital mock-up, but also has defined the rules to ensure that all stakeholders on the job site use the same codes and understand each other. Figure 7 shows in detail all partners involved in Phase 1 of the project, with the particular parcels (lots) they are in charge of. As can be seen, around thirty operators and stakeholders intervene in Phase 1 and provide their own architectural drawings.

Arcadis uses the BIM 360 cloud platform to organize and monitor the district's various components and partners. BIM360 is a cloud-based construction management platform that improves the efficiency and performance of projects. It enables project members to anticipate, optimize, and manage all aspects of project performance through real-time connectivity.

BIM managers have drafted the technical agreement between all of the project's partners to ensure optimal platform utilization. All the district blueprints are stored on the platform, including the following:

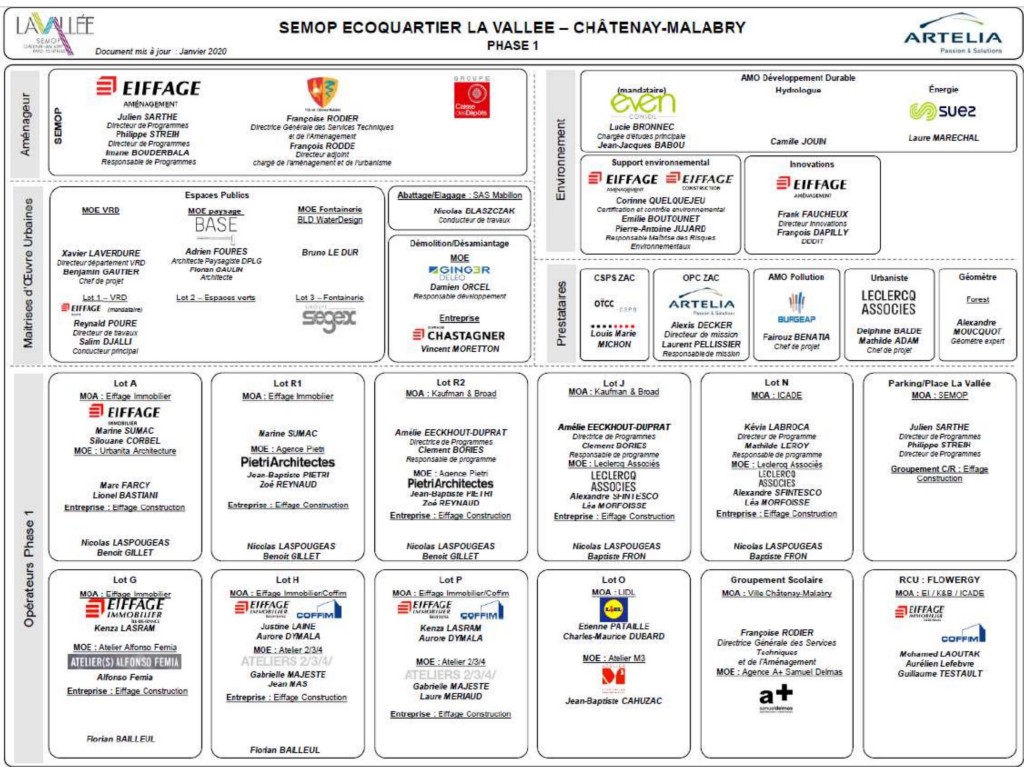

**Figure 7.** AEC partners of project for Phase 1. The real estate project is divided into: planners, urban project management, service providers, and environment and landscape operators.

- The district's general documentation (renamed ZAC);
- 3D general topography, which is the terrain map;
- Urban planning: streetscape including urban furniture and fountains;
- Road network and sidewalks;
- Landscape: green elements of the district, trees, and vegetation;
- Models of real estate parcels not modeled (3D extrusion);
- BIM models of modeled real estate parcels (lots);
- Networks (RSX) classified by job type: gas distribution networks, high-voltage electricity networks, low-voltage electricity networks, drinking water networks, telecommunications networks, public lighting networks, rainwater networks, waste water networks, and heating networks.
- Structures.

With a comprehensive folder system, it is simple to work on multiple areas at the same time while maintaining a general overview of the progress. The technical agreement contains rules that must be adhered to by all parties involved in the development of the project, such as the requirement that they all utilize the same codes and coordinate systems. BIM management has complete access to all of the files, whilst the rest of the partners can only modify, load, and save the folders that are specifically used for their purposes, depending on their field of application.

## 4. Designing the Immersive Visit

### 4.1. Buildings' Repositioning

The data collected from the BIM 360 collaborative platform are first analyzed. The format of the analyzed files is Industry Foundation Classes (IFC), which is a standardized format for the purpose of describing the data in the building and construction sector.

The reference system is RGF93 CC49, and the elevation levels are NGF69. Since 1989, RGF93 has been the French three-dimensional geodetic system using Lambert93 (official projection for maps of France) based on the European ETRS89 system and compatible with

the ITRS global reference system. Thus, the common coordinates enable connecting the owners' digital models. However, analyzing these data, it can be seen that, being made by various architectural firms, the models of the definitive buildings, collected in the last folder in BIM 360, did not respect the same georeferenced coordinates. Each structure had its own Cartesian reference frame.

An IFC file contains all of the building's attributes, from the type of material to the location of each individual point within the structure. This final characteristic enables the examination of the Cartesian system of local reference. A Matlab parser was made to read the IFC files and extrapolate only the data of interest. Along with the parcels, the road infrastructure was assessed, which was divided into three phases (Phase 1, Phase 2, and Phase 3), and the mass plan was assessed as well. The mass plan is a general map of the district.

Because partners of the real estate project failed to choose the same reference point in their BIM, an alignment procedure was required. The map of LaVallée district, along with the absolute coordinates of the different parcels, different roads, and the mass plan, is represented in the following illustration (Figure 8). All the IFC files had to be aligned to the point Ω, manually using Revit, which was time consuming since there are six parcels, one mass plan, and three road networks. Note that just the operation of loading an IFC file under Revit can take up to five to ten minutes. Figure 8 illustrates the resulting model after IFC repositioning.

Figure 9 shows the diagram of the operations required to assemble and reposition the IFC files collected from BIM 360: once joined in a common Revit project, it can be imported into a TwinMotion project using the direct link plugin available under Revit or the Datasmith exporter for more recent versions of Revit.

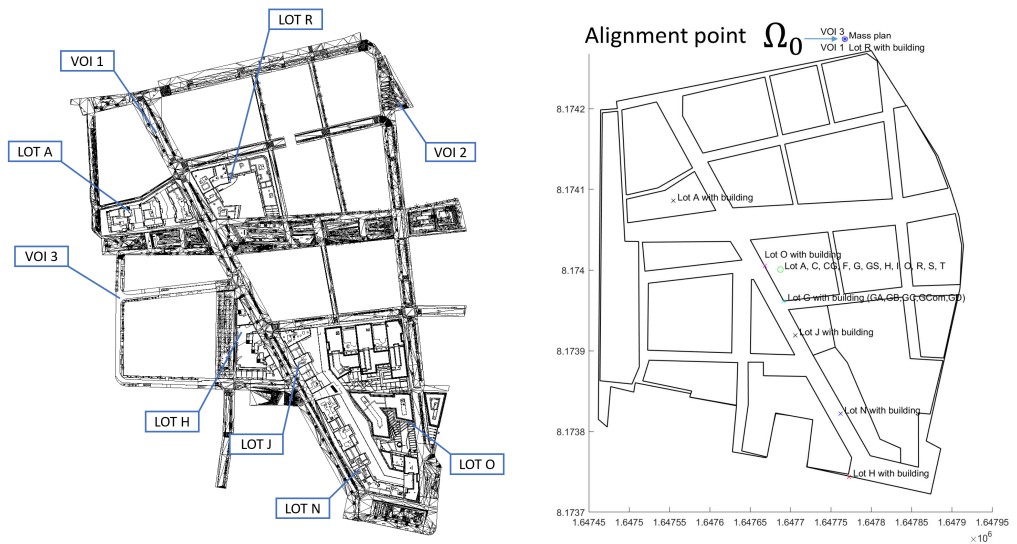

**Figure 8. Left:** Phase 1 of LaVallée on Revit with its parcels (lot), buildings and the road network (VOI). **Right:** absolute reference points of the parcels and the streets.

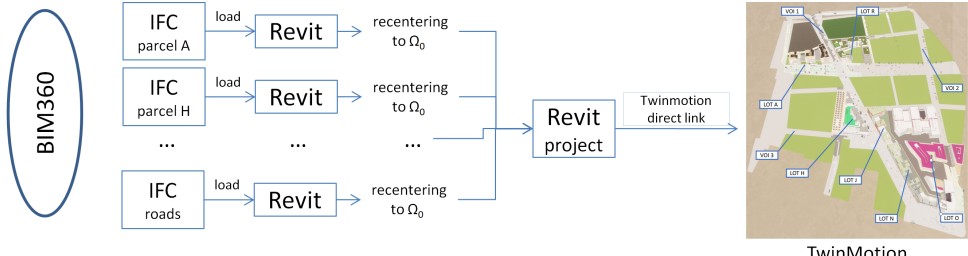

**Figure 9.** Workflow of the construction of the TwinMotion project.

### 4.2. TwinMotion Prototype

A first prototype of the immersive visit was made using archviz TwinMotion, as shown in Figure 10. Phase 1 of the real estate development modeled includes Parcels A, H, N, J, and G, which have been constructed and are illustrated, while Parcel O is now under construction, as well as the road networks, urban furniture, and landscape elements.

After completing the project in Revit and transferring it to TwinMotion, the latter was used to improve the district's characteristics, including material selection and landscape coloring. This application enabled a vivid view of the district's buildings, streets, and urban and landscape furnishings. Therefore, TwinMotion is a particularly useful tool in designing a virtual immersive visit at such a large scale. There are some specificities that one should be aware of: in TwinMotion 2021, the project is very large with 5.2 Go of storage space, but it has good compatibility with the VR headset. With Steam, the demo runs fully in all parts of the district with the HTC Vive pro glasses, with no lags using a high-end PC with an Intel(R) Core(TM) i9-9980HK octocore CPU 2.40GHz, 64 Go RAM, with an NVIDIA Quadro RTX 4000 graphical card. With the same PC and the demo converted under TwinMotion 2022, the project was subjected to a high reduction to 800 Mo: the demo runs in 2D and loads much faster, but the VR HTC Vive headset demo becomes stuck on the still portion of the field of view. Until now, we have not tested it in TwinMotion 2023 and cannot say if this bug has been fixed. Table 1 gives an overview of the principle characteristics of this first prototype of the immersive visit.

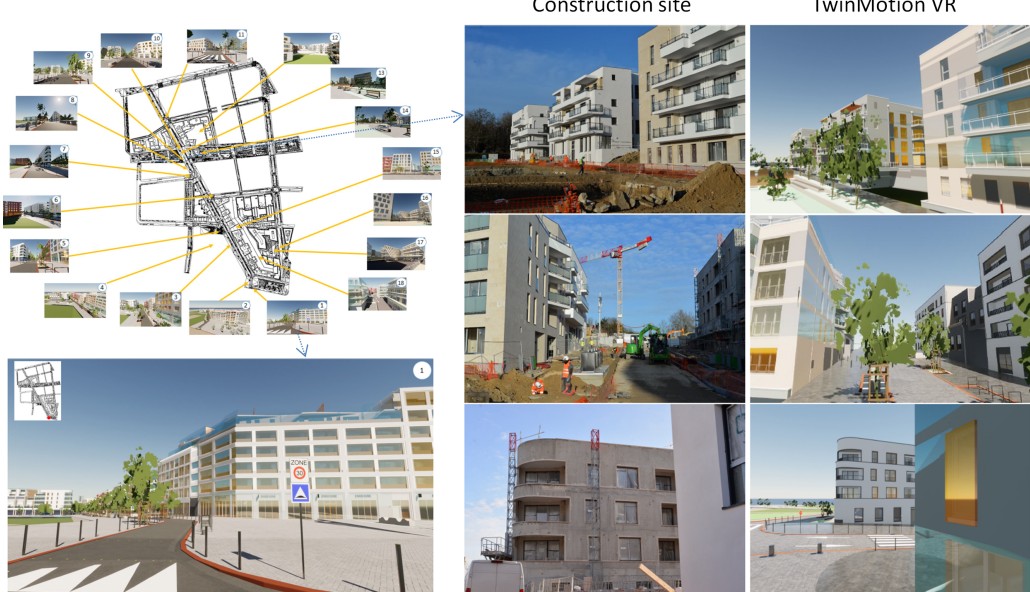

**Figure 10.** TwinMotion visualizations with the reality of the construction site.

**Table 1.** Prototype size and specificity of archviz TwinMotion versions.

| Size of the Project | TM 2020 | TM 2022 |
|---|---|---|
| 24.2 M polygons | 5.2 Go | 800 Mo |
| 77.8 k objects | VR ready | VR not working |
| 0.7 Gb textures | | |

TwinMotion is a powerful architectural visualization software, and it is recommended if one wants to build a fast prototype of a professional 3D city model from BIM files, as it is compatible with Revit. It also has modification capabilities that can be nice for quick corrections. Our only concern was that the resulting buildings and environment were perhaps too colorful, as shown in Figure 10: we compared the resulting buildings and landscape to the reality of the construction site and found it too *cartoonish*, which can be nice for urban planning purposes and a quick demonstration, but can introduce a bias if

one is looking for a more photorealistic immersive experience. Another problem is that if we wanted to add dynamic features to the scene such as moving persons or cars following a precalculated or real-time scenario, which is not possible with TwinMotion. Therefore, we decided to investigate the further possibilities brought by Unreal Engine.

### 4.3. Manual Work and Theoretical Elements in Unreal Engine

In this part, we see what can be manually done to improve the architectural model in TwinMotion and in Unreal Engine. Many features used in Unreal have some equivalent in TwinMotion, but the impossibility of using scripting in TwinMotion becomes the main argument to use Unreal Engine instead for every step detailed below.

### 4.3.1. Improvement with TwinMotion

The source files of our models have the .ifc extension. Those files can be opened in Revit, and then, thanks to a Unreal Datasmith Plugin, the resulting Revit file can be imported into TwinMotion. The model used in Unreal Engine comes from the TwinMotion prototype.

At this stage, two operations can be massively performed manually under TwinMotion to enhance the prototype, in order to place the 3D model in its city and the placement of more realistic vegetation:

- Import the urban context: The urban context of a project is a square zone selected from a world map, from which the software extracts a rough 3D model of the surrounding city, which corresponds to a group of white parallelepipeds placed above a gray plane. A context is useful to fill up spaces in an architectural project and then give a better impression of realism. Figure 11 illustrates the urban context extracted for this project.
- Manual tree replacement: For each species of tree, we have to find an equivalent in TwinMotion, either by species or by shape. Nevertheless, in order not to weigh down the file of the virtual tour any further, we may not replace all the trees with their estimated equivalents, performed following Table A1 of Appendix A. The equivalences by shape was performed manually, by checking the settings of each TwinMotion tree's form (age, height, and season) and then personalizing them accordingly; see Figure A1.

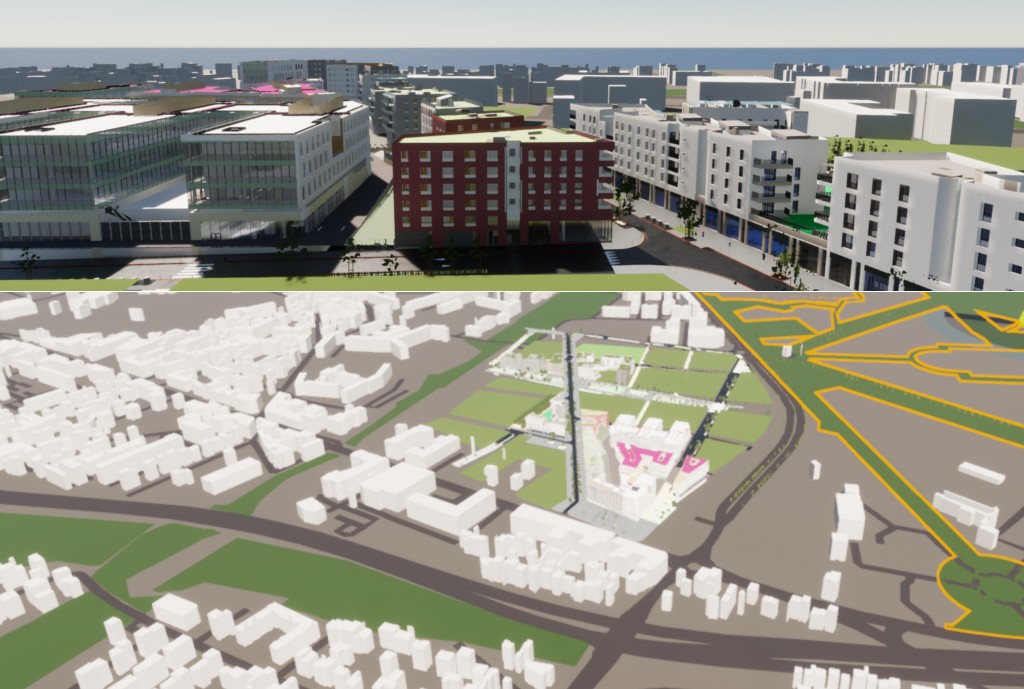

**Figure 11.** LaVallée model with urban context from TwinMotion.

The resulting prototype in TwinMotion was the one used for further improvements in Unreal Engine (UE).

### 4.3.2. Import of the 3D models in the game engine

With Unreal Engine, importing an FBX or OBJ file is simple and fast. Nevertheless, the projects we are working on use 3D models from architecture-oriented software, such as Revit or Naviswork, which requires additional various steps to achieve a decent result. In this part, we discuss how to import the various file extensions that we can import into Unreal, the methods that can be used to do this, and then, the small possible operations to fix some import problems.

Figure 12 illustrates the main source files that can be imported for architectural data handling:

- IFC .ifc files and CAD files can be imported directly by Unreal.
- TwinMotion project .tm files can be imported by using Datasmith if the TwinMotion plugins are enabled.
- Other files such as Naviswork or Revit files (.nwd, .rvt) should be converted to the .udatasmith format first.

Therefore, specific plugins have to be enabled, as represented in Figure 12. The Python editor script plugin also has to be selected for automation purposes.

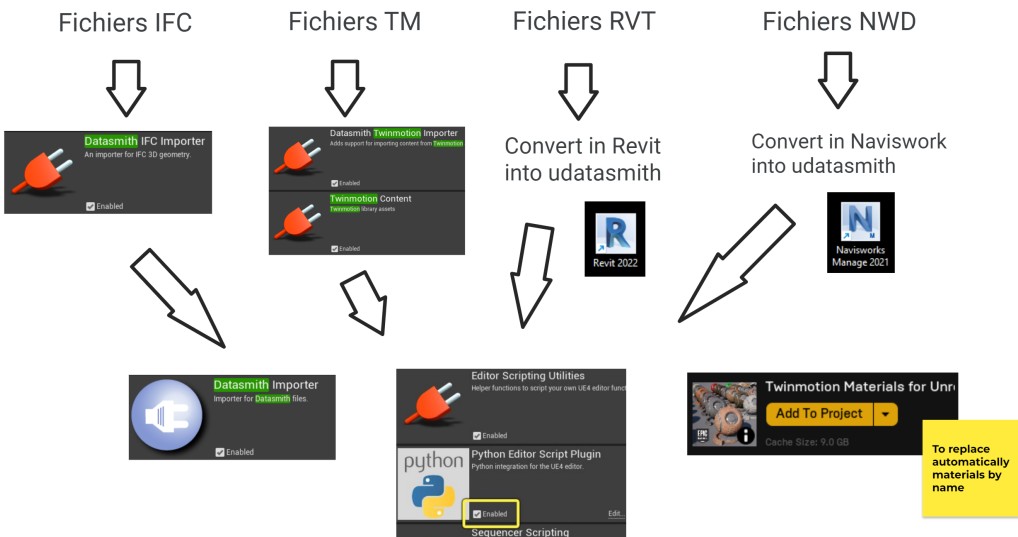

**Figure 12.** Examples of workflow with the Datasmith exporter plugin.

Some limits of imports in general are as follows:

- When importing several 3D models, one may want them to be in a different location than the one in their source file. Neither Datasmith nor Dataprep give us that option, and then we should manually move the highest parent actor of our imported model to move the whole model correctly.
- When creating Unreal assets equivalent to the objects of the source file, for some plugins, some objects (such as cameras or light objects) are not taken into account or can be badly created. For instance, the materials created may be visually totally different from the ones used in the source file (it is possible to have blank materials instead of realistic ones). In this case, we need to develop a way to automatically change the materials, which is explained below.
- The Unreal Engine plugins seek to create a hierarchy that resembles the source file as much as possible. Knowing that, it is sometimes necessary to be careful with the model in some software, in order not to obtain an over-weighted file. For instance, when importing a 3D model from Sketchup, the model of a tree may be imported in

Unreal by creating an actor for each mesh of a leaf (which is resource consuming and can cause some visual problems in the unreal project).

### 4.4. Creation of an Automated Pipeline in Unreal Engine

TwinMotion is not a 3D modeling software, nor a software that enables scripting. To solve some of those issues, we intend to develop tools from Unreal Engine that would enable more options. By using scripting, for instance, we can automatize the replacement of a group of buildings or program micro-mobility movements.

The choice of using Unreal Engine was made by knowing that TwinMotion and Unreal were both developed by EpicGame, so we expected an easier setup of the various tools in Unreal.

#### 4.4.1. How to Use Unreal Python Scripting

Setting up a virtual tour of LaVallée requires knowing some features of Unreal that may help make the rendering more efficient, the plugins to download, making a specific headset usable, and the possible ways to experience the visit in immersive VR (one can check the VR performance features in the documentation with the link: https://docs.unrealengine.com/4.26/en-US/SharingAndReleasing/XRDevelopment/VR, accessed on 22 February 2023).

After setting up the project appropriately, one can then use Unreal Python scripting to make the process of importing and massively modifying specific elements in the model faster. This section presents a short introduction on how one can use scripting to automatize most of the steps that are usually performed manually by practitioners. It also explains which Unreal tools can be used for which purposes and how to use the Unreal scripting documentation to find if an idea is doable with Unreal available assets or functions.

What are not detailed in this part are all the different ways to achieve the same results and all the prototype algorithms tested during our experimentation. The following methods tend to be more efficient or easy-to-use than some brute-force methods that do not use certain existing functions or assets.

Installations before Executing Script

- Enable the *Editor Scripting Utilities* plugin
  (*Python Editor Script Plugin* is enabled by default);
- Download the Python scripts and config files shared alongside this article.

There are various ways to execute Python scripts: the main ways one can use a Python file or chunk of code are illustrated in Figure 13.

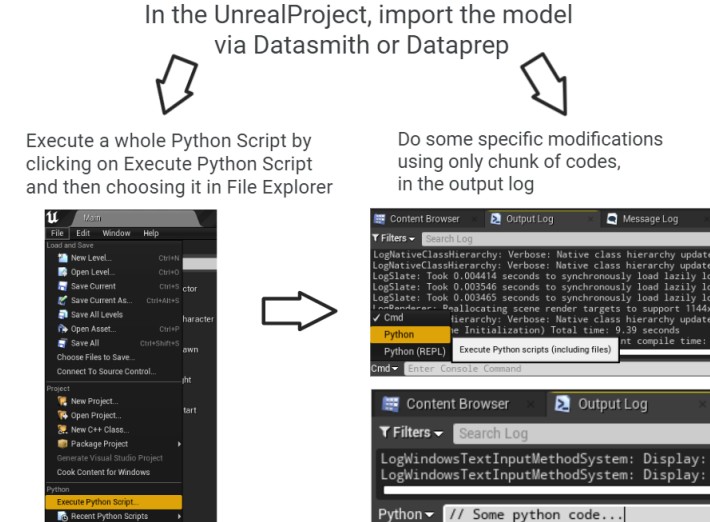

**Figure 13.** After import, execute Python script from files or using OutputLog.

The Unreal Python scripting documentation is available at the following address and is an important source of information: https://docs.unrealengine.com/4.27/en-US/PythonAPI/, https://docs.unrealengine.com/4.27/en-US/PythonAPI/, accessed on 22 February 2023.

The pieces of script we worked on are partly visible in the explanations below. The whole scripts are in the Jupiter Notebook returned alongside this paper as additional material.

Generally useful Unreal classes and functions: People who use Python scripting generally work on the persistent-level actors. The most-obvious functions one may think of are those that enable spawning/deleting actors, selecting some actors, selecting some assets found in the content browser, etc. Figure A2 shows an extract of the most-useful functions needed for basic interactions with actors that exist at the persistent level or with available assets that one wants to spawn at that level.

Many of these functions help write a simple script faster. For instance, when replacing the material of an actor, one needs to know how to access the reference of the actor's material list. Actually, one has to change the material from its MeshComponent (and not from its StaticMesh...). Instead of writing every little step to access and change the material for each actor of a list, it is easier to know the existence of the function (cf. Figure A2): *unreal.EditorLevelLibrary.replace_mesh_components_materials_on_actors*(...)

How to read the Unreal documentation of this paper: the form of the documentation that we chose needs to be explained, although it is straightforward. We did this manually with the following conventions:

- Each gray rectangle represents an Unreal class;
- At the top of a rectangle, there are the class name and, sometimes, a legend;
- Below this, there are the class functions or properties:
  - If the line begins with a dot, this means that one can use it directly by calling it from the class, as in *unreal.XClassName.function*(...) or in *unreal.XClassName.Xproperty = ValueOfCorrectType*;
  - If there is no dot, this means that this is a property that is used by calling it within the functions that obtain or set the editor property, e.g., *unreal.XClassName.get_editor_property*(*property_name*) or in *unreal.XClassName.set_editor_property*(*property_name, ValueOfCorrectType*);
- A thin-line arrow helps to indicate the type of a property or function's result. As an example: *.function $\longrightarrow$ unreal.XClassName*[*...description...*];
- A thick arrow helps to indicate a class's hierarchy. In other words: *unreal.ClassA $\Longrightarrow$* class that inherits or implements *ClassA*
- A line beginning with "//" is a comment;
- The tables of functions can be read intuitively.

### 4.4.2. Automatic Loading of the 3D Models with Datasmith

For the pipeline, we decided to directly import the 3D models of the TwinMotion prototype with Datasmith. There are other Datasmith-related Unreal classes, as illustrated by Figure A3 of Appendix B, but those are apparently not relevant to import the models. To do so simply, one only needs to construct a DatasmithScene from the file that contains the model and then import it. The corresponding code can be found in Appendix B. The most-important functions are the one that loads a TM project and the one that removes an element of the loaded scene.

```
ds_file_on_disk = r"C:\Users\Usuario\MyTwinMotionProject.tm"
ds_scene_in_memory = unreal.DatasmithSceneElement.
    construct_datasmith_scene_from_file(ds_file_on_disk)
```

For our 3D models, even though the useless elements can be deleted after the import, we chose to do this beforehand, which can save a bit of the memory space when auto-saving. For this purpose, there is the function:

```
unreal.DatasmithSceneElement.remove_mesh(mesh)
```

### 4.4.3. Some Generalities about Actors and Their Components

Some functions to replace some actor's materials or meshes can be found, but one would not know about how this is basically performed (in the most-similar way to the manual one). Figure 14 helps visualize what is happening.

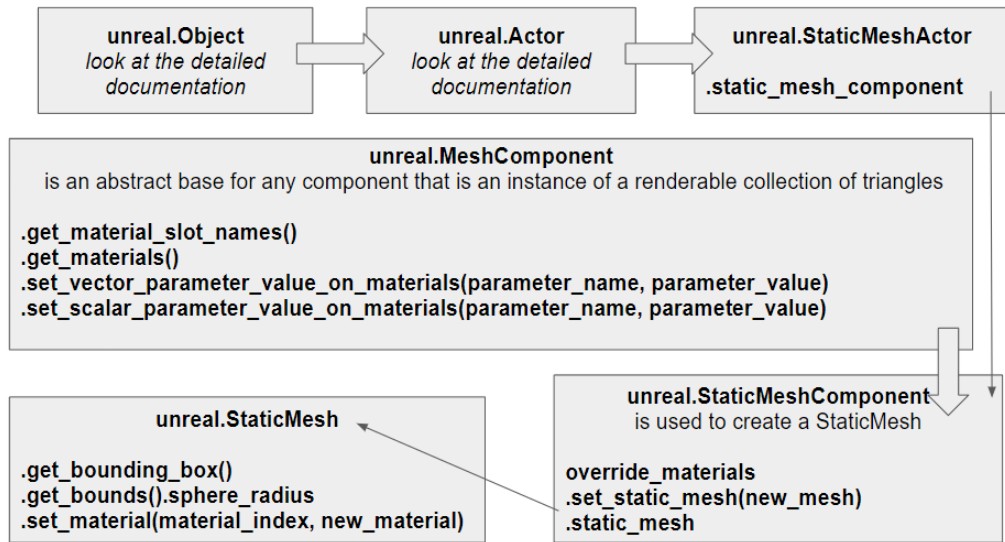

**Figure 14.** Unreal documentation: some Unreal classes that are useful to know when performing material and mesh replacement.

When importing a mesh in Unreal visibly in the viewport, one can see it as a StaticMesh, contained in a StaticMeshComponent, contained in a StaticMeshActor. The StaticMesh-Component is used so that it can override the material of the imported asset (which enables having actors with the same mesh, but different materials) instead of changing the Static-Material of the asset (which could change the material of actors using the same mesh or an invisible result in the viewport).

### 4.4.4. Automating Material Replacement

At this stage of the UE prototype, there is the necessity to replace the material. The created materials, during an import from TwinMotion files or others, can often be blank and flat. Instead, we need to replace those default materials with realistic ones such as the one available in UE illustrated in Figure 15.

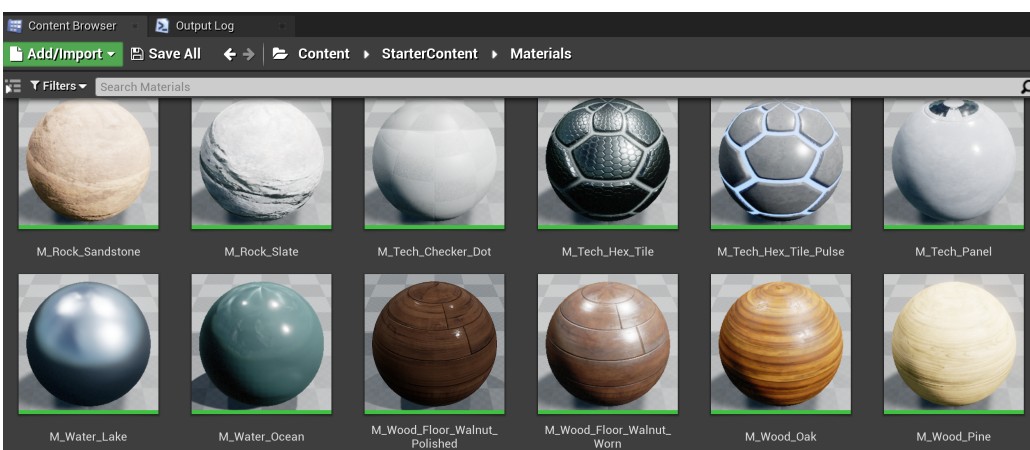

**Figure 15.** Example of realistic materials available in Unreal Engine.

Using the assets of existing plugins seems to be the easiest and fastest way. The TwinMotionContent plugin would be enough with its large panel of assets, including

materials. Yet, if we want to only change materials with some similar TwinMotion material, we may use a smaller plugin such as TwinMotionMaterial (which already weighs around 9 Go). The material assets of the plugins are MaterialInstances.

How to determine material equivalences: we can expect the names of the original unrealistic materials to be explicit enough for a person to understand their equivalents in real life. For instance, if a material is called *"old_copper"*, one can easily find a similar realistic material and replace it manually.

To avoid doing everything manually, we intuitively want to create tables with two columns: one for strings *s* to search in a material name and another for a realistic material reference that would replace the material that contains string *s*. This is exactly what a DataTable is used for. A DataTable is a grouping of records that all share common properties, which can then be accessed by row references or row keys. In other words, it is a simple key value store.

One MaterialSubstitutionDataTable can be created manually by adding rows (a string to search + a replacement material). When using a Dataprep function to use the DataTable, the replacement is performed by checking the name of each filtered actor by reading each row in order: the original material can be replaced according to the first row, but then, also, according to next rows, etc.

Fast material replacement can be performing using the DataTables and Dataprep functions. Dataprep assets provide reusable recipes that help with being consistent about how to import and modify the 3D data; see Figure A4 in Appendix C. Those recipes are created to reorganize, clean, merge, and modify scene elements before creating the final assets and actors in an Unreal Engine project.

Assets for material replacement: Instead of a .csv file, we preferred using a more efficient tool commonly used when importing models with Dataprep. We used several MaterialSubstitutionDataTables.

It is possible to use only one sufficiently general to be used in the first place in any file (replace the materials of type "steel" with the realistic materials of type "steel", and so on for all materials). Yet, we used several DataTables in order to be more precise in the replacement. For each mesh, we can try using wisely all the DataTables or using specific DataTables chosen according to the name of the mesh.

How to be more precise in the replacement: Instead of being satisfied with a general DataTable, which would only contain the general ideas of the materials, we wanted to be more specific. For instance, using several DataTables enabled us to be precise if the replacement metallic-like material should be old looking or not.

The set of DataTables was created using the references of the assets in TwinMotionMaterial plugin. In the plugin, the materials are classified by the following looking characteristics: *brick, ceiling, concrete, fabric, glass, grid, ground, leather, marble, metal, modeling, parquet, plastic, roof, stone, tile, wall*, which explains the structures of our created DataTables:
Method 1: Material replacement with one DataTable;
Method 2: Material replacement with succession of DataTables;
Method 3: Material replacement with specific DataTables for each mesh; Method 3 is described with its Python code in Appendix D.

### 4.4.5. Automating Vegetation Mesh Replacement

Vegetation mesh replacement can be performed using a .csv file and Dataprep functions. Assets for mesh replacement: It would have been possible to create a new form of DataTable for mesh substitution, but the structure would be such that we would have to neglect the age of the vegetation. We preferred to use a **.csv file** with the available tree names, then indicate the reference according to the age desired by the architect.

Method 1: Replace the meshes of several actors, using a table with a minimum of two columns (*meshes_to_search*, *loaded_meshes*) and the Dataprep library:

```
def replace_meshes(selected_objs,meshes_search,mesh_substitutes):
 string_match = unreal.EditorScriptingStringMatchType.CONTAINS
 for m_in,m_out in zip(meshes_search,mesh_substitutes) :
```

```
unreal.DataprepOperationsLibrary.
substitute_mesh(selected_objs, m_in, string_match, m_out)
```

Nevertheless, instead of using simple table, one wants to take into account the age of the tree: when the tree is a "Tige" or an "Arbuste", one wants to be able to choose the youngest version from a tree mesh. This means that one would like to create the mesh replacement table by looking at two keywords in the name of a mesh (the tree species and an age indicator), which leads us to the second method.

Method 2: Replace meshes by directly setting the MeshComponent of the actor: There are three StaticMeshes of a tree species: a young ($SM\_Y$), a middle-aged ($SM\_M$), and an aged ($SM\_A$) version. Most of the time, one would want to access the middle-aged trees.

To access those meshes, directly loading them with the absolute path will not work. One must enable the TwinMotionContent plugin in the project and then use the path of the plugin that Unreal understands:

"$/TwinMotionToUnrealContent/Library/Vegetation/Trees/$"

The Python code for vegetation replacement is given in Appendix A.

*Nota Bene*: To load an asset located somewhere in the content folder of an Unreal project, the path to use begins with "$/Game/$" and uses the directory from the content, as if the content does not exist. To load an asset located in a plugin, write a path similar to "$/PluginName/RemainingDirAsIfContentDoesNotExist$".

With such a method, an extract of the result can be seen in Figure 16.

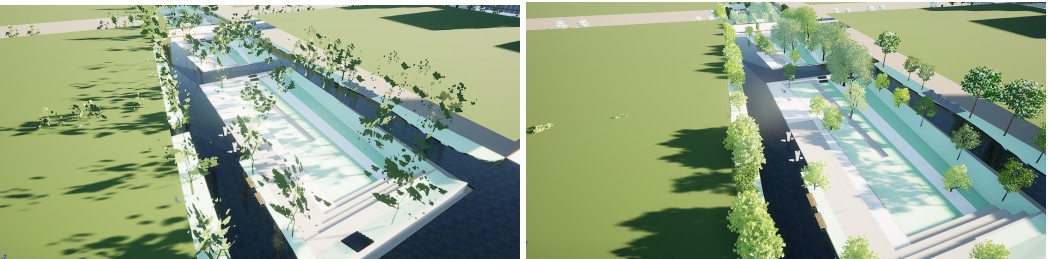

**Figure 16.** Comparison before and after the vegetation mesh replacement.

### 4.4.6. Automating Light Settings

The parameters we want to set are illustrated in Figure 17. To do so, we do not want to directly set all the features in the light objects, because some parameters are only accessible from their LightComponent. Thus, for the light objects we want to use, we first obtain their LightComponent, which can be of various classes (unreal.LightComponent, unreal.SkyLightComponent, etc). See the Python code extract in Appendix E for a further explanation.

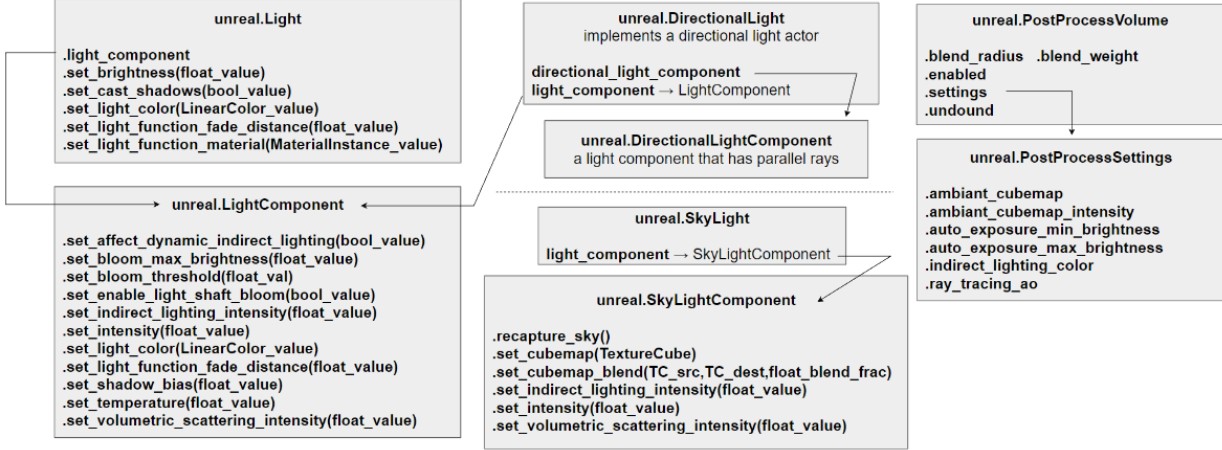

**Figure 17.** Unreal documentation: Unreal classes for lighting.

In addition to setting the parameters mentioned, do not forget to correctly set the mobility of those objects. In particular, when using MeshDistanceField, it appears preferable to set SkyLight to *Stationary* instead of to *Movable*.

Figure 18 illustrates the types of modifications for lighting for a scene extracted from our district of interest.

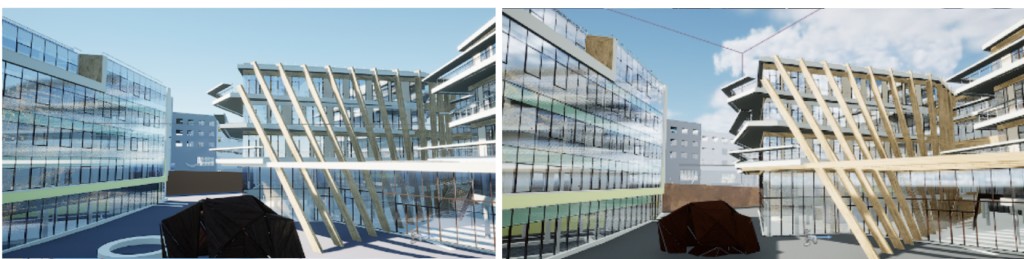

**Figure 18.** Before/after light settings.

As a matter of simplicity, in the extract of the code detailed in Appendix E, some assets' settings are not shown, such as VolumetricClouds, ExponentialHeightFog, SphereReflectionCapture, etc. To see the whole script (with the whole settings), check the Jupiter Notebook given alongside this article.

**NB:** Automatizing the lighting is not as necessary as automatizing the replacement of the materials and vegetation meshes. It is possible to make all of the settings manually, but it should be at least checked manually to achieve the best photo-realistic view possible.

## 5. Results

Let us remind that the goal of our immersive visit was to make a platform on which architectural studies can be produced. Therefore, the realism of the obtained visit is essential to the project. Figure 19 illustrates the kind of improvements our method can be bring compared to a standard TwinMotion project.

One might argue that such a result could be obtained with TwinMotion, but it would be at the cost of many manual operations. Indeed, the city model of Phase 1 of LaVallée contains 34 buildings, road networks, urban furniture, and landscape vegetation, which would require a modification at least on the shell of the building. Furthermore, the integration of the Phase 2 and Phase 3 models that will be collected from the architectural firms can now be automatically modified: textures, materials, lighting, and vegetation are already set up, just by running the Unreal Engine Python scripting. Furthermore, TwinMotion does not allow one to integrate one's own simulation results such as the motion of people, bicycles, and cars or other type of results, for instance physical flows such as computer fluid dynamics.

Figure 20 illustrates the result of Phase 1. As can be seen, various atmospheres are present in this 3D model of the future district. They represent the various kinds of activities that will be available to the public. A green leisure area in the shape of a pedestrian way can be used to assess the effect of vegetation and soft modes of transportation on well-being. A large shopping strip crosses the entire district. Office space, as well as retail space are also available, not to mention that the insides of the buildings are also visit-able, even though we did not illustrate them in this paper.

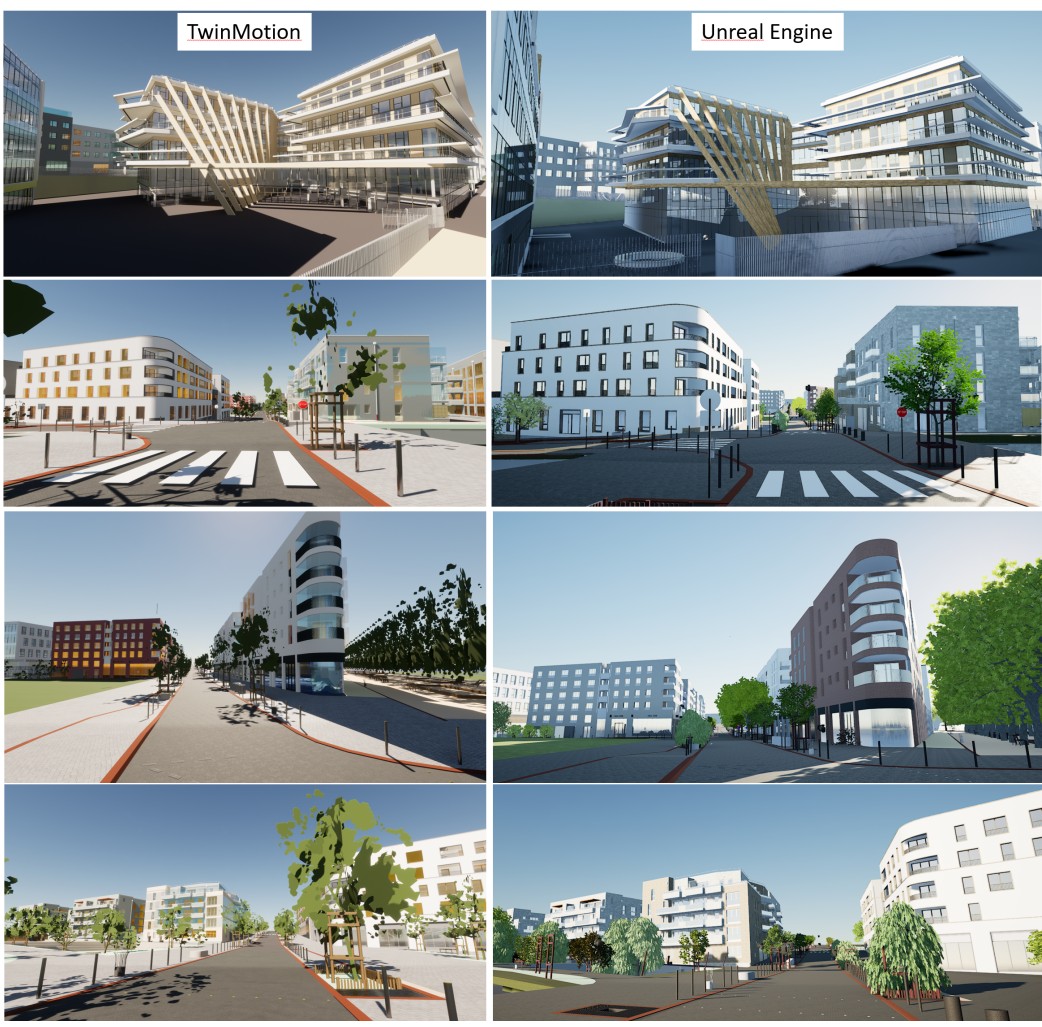

**Figure 19.** Left: TwinMotion project. Right: Unreal Engine scene. The Unreal Engine texture and lighting conform more to reality than the TwinMotion scene, which is more cartoonish.

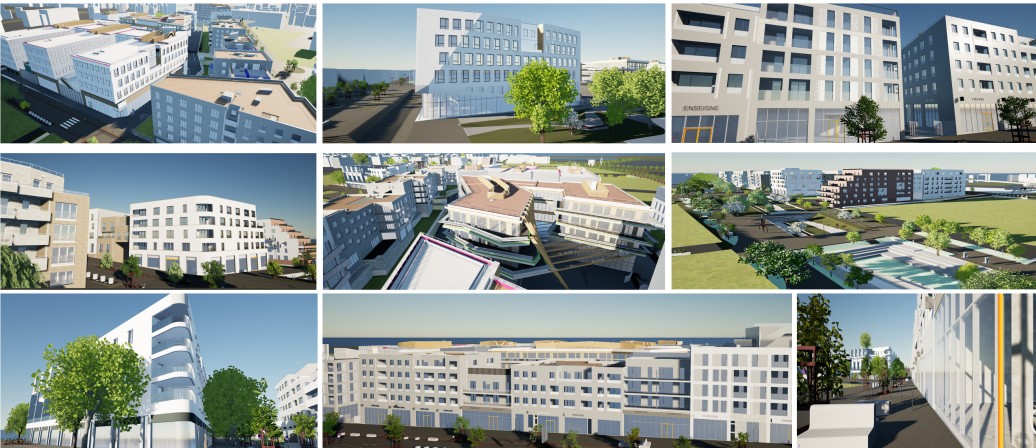

**Figure 20.** Gallery of points of view of the final immersive visit: many various architectural environments are represented.

## 6. Discussion

This work was the result of a partnership between a real estate developer, various architectural firms, a BIM management company, and the academy; this program intends to use a real estate project under construction as an opportunity to experiment with new

research concepts in the design of urban environments. The industrial operators were the providers of all the BIMs used to build our immersive visit of LaVallée: buildings, roads, landscape, and urban furniture.

The main technical development of this work consisted of assembling these 3D models into one aggregated large-scale model and enhancing it to make it more realistic. This research was split into a technical part and a research part, which mainly consisted of developing possible methodologies for the experiments on architectural environments with VR tools. To develop the virtual tour, it was necessary to obtain a more realistic model. To do so with Unreal Engine, several tasks can be carried out manually in the case of a small model, but due to the large scale of our model, we proposed several ways of developing the tools to speed up those steps and automate them.

During this work, we created various sets of Unreal assets and Python scripts to perform the stated tasks much more simply and quickly. For the research study, we created more realistic scenes extracted from the global model and improved the global model using the pipeline implemented with Unreal Python scripting. For the automation of realism creation process, we built DataTables associated with the assets of the TwinMotionMaterials plugin and devised scripts related to each step of the whole process. For more technical details, a Jupiter Notebook is provided as additional material on how to use the assets and how to manage some potential problems.

Various improvements can be highlighted, such as finding a way to automatize the correct UV mapping or auto-scaling of our material assets, to avoid any strange texture projections. We are in the process of adding micromobility elements to make our virtual scenes dynamic, with people, bicycles, and cars moving at the scale of the district. We will improve the road network model file by creating a more accurate model and using various altitudes, which will allows us to think about car spawning strategies.

Optionally, will export the work in an APK file to be able to put the model on Android for use with VR headsets that allow these kinds of features, such as the HTC Vive Focus, enabling us to perform VR experiments without wires.

As for the output of the presented project, we are in the process of using the developed immersive visit of LaVallée in VR study related to architectural environments. We would like to set up experiments that would bring some answers to some specific questions. The research subject is not yet definitively defined, but would likely focus on one of the following points such as: Can VR be or become a tool useful to teach the ability to predict areas of interest (hot spots) in an urbanism project as large as the LaVallée project? How can the differences of perception between 2D and 3D be characterized. Other questions relate to the potential differences of ambiance perception when using an indoor point of view or an outdoor one of a virtual environment or the influence of photo-realism in the perception of ambiance.

**Author Contributions:** Conceptualization, R.B. and J.M.G.; methodology, H.D., R.B. and J.M.G.; software, R.B. and H.D.; validation, E.D.G. and H.D.; formal analysis, R.B. and H.D.; investigation, R.B. and H.D.; resources, R.B. and J.M.G.; data curation, H.D.; writing—original draft preparation, R.B. and H.D.; writing—review and editing, E.D.G. and J.M.G.; visualization, H.D. and R.B.; supervision, R.B., E.D.G. and J.M.G.; project administration, R.B.; funding acquisition, R.B. All authors have read and agreed to the published version of the manuscript.

**Funding:** This research was funded by the E3S project, a partnership between Eiffage and the I-SITE FUTURE consortium. FUTURE bénéficie d'une aide de l'État gérée par l'Agence Nationale de la Recherche (ANR) au titre du programme d'Investissements d'Avenir (référence ANR-16-IDEX-0003) en complément des apports des établissements et partenaires impliqués.

**Institutional Review Board Statement:** Not applicable.

**Informed Consent Statement:** Not applicable.

**Acknowledgments:** The authors would like to thank Arcadis company for providing the assembled IFC files of the building model of each parcel.

**Conflicts of Interest:** The authors declare no conflict of interest. The funders had no role in the design of the study; in the collection, analyses, or interpretation of the data; in the writing of the manuscript; nor in the decision to publish the results.

## Appendix A. Vegetation-Related Elements

**Table A1.** Table of tree mesh equivalences.

| Imported from IFC | TwinMotion Tree |
| --- | --- |
| Acer | Vine maple |
| Amelanchier | Holm oak (by shape) |
| Betula | Sweet birch, grey birch |
| Carpinus | Persian ironwood (by shape) |
| Fraxinus | Manna ash |
| Liquidambar | Sweet gum (young + winter mode) |
| Malus | Apple tree |
| Parrotia | Persian ironwood |
| Prunus | Sweet cherry tree, peach tree |
| Pyrus | Pear tree |
| Quercus | Cork oak |
| Salix | Weeping willow |
| Tilia | Littleleaf linden, American linden |
| Ulmus | Littleleaf linden (by shape) |
| Zelkova | Laurel (by shape) |

Mesh replacement code:

```python
tags = ["EV_Arb","ARB","Arbre","Tree"]
default_tree_name = "_QuercusRubra"

# same as you if you used default download settings of the plugin:
trees_path= "/TwinMotionToUnrealContent/Library/Vegetation/Trees/"
# parsing csv table into table of string with mesh names :
table_file_path = r"C:\Users\Usuario\mesh_replacements.csv"
mesh_table = csv_into_table(table_file_path,",","\n","\"")

selected_objs = get_all_actors_with_tags(tags)
selected_objs = unreal.DataprepFilterLibrary.filter_by_class(
    selected_objs, unreal.StaticMeshActor)
for a in selected_objs :
  matches = [x[1] for x in mesh_table
      if x[0] in a.get_actor_label()]
  # default tree name if no matches
  name = default_tree_name
  if len(matches) > 0 :
    name = matches[0]
  # full name is ref of the tree
  full_name = trees_path + "XF" + name + "/"
  if "Tige" in a.get_actor_label() or "Arbuste" in a.get_actor_label() :
    full_name += "SM_Y"
  else :
    full_name += "SM_M"
  full_name += name +"_LOD0"
  # replace mesh using MeshComponent
  mc = a.get_component_by_class(unreal.MeshComponent)
  sm = unreal.load_asset(full_name)
  if sm != None :
```

```
mc.set_static_mesh(sm)
# set a random Z rotation to appear less redundant
mc.set_editor_property("relative_rotation",unreal.Rotator(0,0,randint(0,
                            181)))
# reset overriding materials, to get right ones
mc.set_editor_property("override_materials",sm.static_materials)
```

<div align="center">

Carpinus betulus         Persian Ironwood

</div>

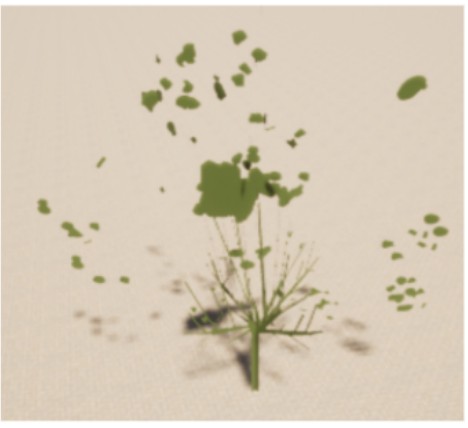 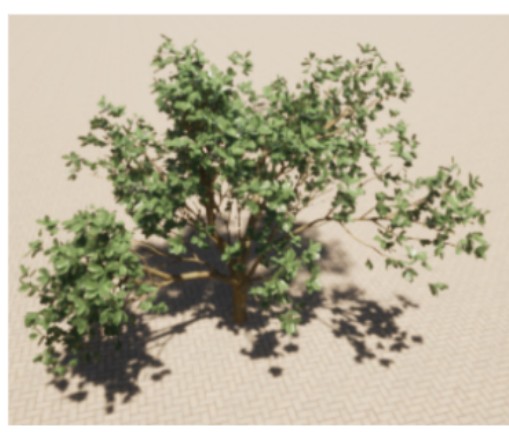

**Figure A1.** Example of tree equivalence by shape.

---

**unreal.EditorAssetLibrary**

```
.checkout_asset(asset_to_checkout)
.do_assets_exist(asset_paths)
.list_asset_by_tag_value(tag_name, tag_value)
.load_asset(asset_path)
.rename_asset(source_asset_path, destination_asset_path)
.save_asset(asset_to_save, only_if_is_dirty=True)
```

---

**unreal.EditorStaticMeshLibrary**

```
.add_simple_collisions(static_mesh, shape_type)
.set_convex_decomposition_collisions(static_mesh, hull_count, max_hull_verts, hull_precision)
.add_uv_channel(static_mesh, lod_index)
.generate_box_uv_channel(static_mesh, lod_index, uv_channel_index, position, orientation, size)
.set_generate_lightmap_uv(static_mesh, generate_lightmap_u_vs)
```

---

**unreal.EditorLevelLibrary**

```
.clear_actor_selection_set()
.destroy_actor(actor_to_destroy)
.get_all_level_actors()
.replace_mesh_components_materials_on_actors(actors, material_to_be_replaced, new_material)
.replace_mesh_components_meshes(mesh_components, mesh_to_be_replaced, new_mesh)
.replace_mesh_components_meshes_on_actors(actors, mesh_to_be_replaced, new_mesh)
.replace_selected_actors(asset_path)
.save_current_level()
.set_actor_selection_state(actor, should_be_selected)
.set_selected_level_actors(actors_to_select)
.spawn_actor_from_class(actor_class, location, rotation=[0.0, 0.0, 0.0], transient=False)
.spawn_actor_from_object(object_to_use, location, rotation=[0.0, 0.0, 0.0], transient=False)
```

**Figure A2.** Unreal documentation: some generally useful functions.

## Appendix B. Datasmith Functions and Program to Load a TwinMotion Project

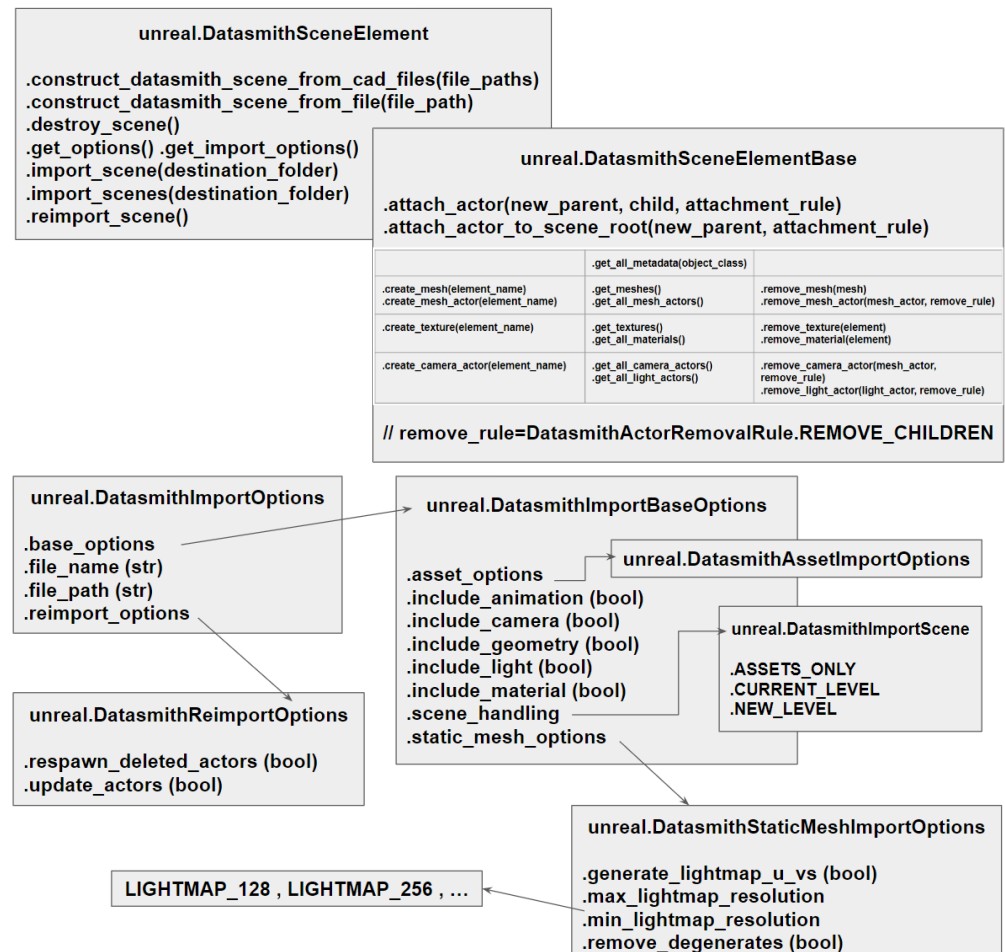

**Figure A3.** Unreal documentation: Datasmith-related Unreal classes.

```python
ds_file_on_disk = r"C:\Users\Usuario\Documents\TestImport.tm"
ds_scene_in_memory = unreal.DatasmithSceneElement.
    construct_datasmith_scene_from_file(ds_file_on_disk)

if ds_scene_in_memory is None:
 print("Scene loading failed.")
 quit()
#-------------------------------------------------------------
# Remove any mesh whose name includes certain keywords
#-------------------------------------------------------------
ifc_rmv_kw = ["IfcSpace","IfcOpeningElements"]
rwd_rmv_kw = ["3D","Pieces"]
tm_rmv_kw = ["RPC_Femme","RPC_Homme"]

# remove_keywords = list that depends on file extension [...]
meshes_to_skip = [] # set of mesh actors whose name is in rmv_kw

# Remove all the meshes we don't need to import :
for mesh in meshes_to_skip:
 ds_scene_in_memory.remove_mesh(mesh)

# Set import options [...]

# Destination folder must start with /Game/ :
result = ds_scene_in_memory.import_scene("/Game/MyImportedScenes")

# For our TwinMotion file created from an IFC file, delete Actors
```

```
# whose 1st material name is "IfcSpace", "IfcOpeningElement" [...]

#----------------------------------------------------------------
# Finish import
#----------------------------------------------------------------
if not result.import_succeed:
 print("Importing failed.")
 quit()

# Clean up the Datasmith Scene.
ds_scene_in_memory.destroy_scene()
```

## Appendix C. Dataprep-Related Elements

---

**unreal.DataprepOperationsLibrary**

.substitute_material(selected_objects, material_search, string_match, material_substitute)
.substitute_materials_by_table(selected_objects, data_table)
.substitute_mesh(selected_objects, mesh_search, string_match, mesh_substitute)
.randomize_transform(selected_objects, transform_type, reference_frame, min, max)
.set_simple_collision(selected_objects, shape_type)
.set_convex_decomposition_collision(selected_objects, hull_count, max_hull_verts, hull_precision)

// string_match = 0 means "contains" , 2 means "exact match"

---

**unreal.DataprepFilterLibrary**

.filter_by_class(target_array, object_class)
.filter_by_name(target_array, name_sub_string, string_match)
.filter_by_size(target_array, size_source, filter_mode, threshold)
.filter_by_tag(target_array, tag)

---

**Figure A4.** Unreal documentation: Dataprep useful functions.

## Appendix D. Python Code for Material Replacement

```
#----------------------------------------------------------------
# METHOD 3 : Material replacement with specific DT for each mesh
#----------------------------------------------------------------

selected_objs = unreal.EditorLevelLibrary.get_all_level_actors()
# selected_objs filtered by name or by class [...]
 # unreal.DataprepFilterLibrary.filter_by_class(...)
 # unreal.DataprepFilterLibrary.filter_by_name(...)

# Array that track unchanged Actors :
untouched_objects = unreal.Array(unreal.Object)
# List of keywords associated to each DataTable
DTofDT_keys = unreal.StringTableLibrary.get_keys_from_string_table
  ('/Game/Tests_lib/DataTables/Table_of_DT.Table_of_DT')

# Determine and use appropriate DT for each obj :
for obj in selected_objs :
 obj_mcomp = obj.get_component_by_class(unreal.MeshComponent)
 obj_mat_prev = obj_mcomp.get_materials()
 for key in DTofDT_keys :
  if obj.get_name() in key :
   value = unreal.StringTableLibrary.
     get_table_entry_source_string('/Game/Tests_lib/
     DataTables/Table_of_DT.Table_of_DT',key)
   DT = unreal.load_asset(key)
   # Using one DT on one obj :
   unreal.DataprepOperationsLibrary.
     substitute_materials_by_table([obj], DT)
 obj_mat_next = obj_mcomp.get_materials()
 if obj_mat_prev == obj_mat_next :
  untouched_objects.append(obj)
# For untouched_objects, you can e.g. use Method 2 [...]
```

## Appendix E. Setting Up the Lighting Automatically

```python
#---------------------------------------------------------------
# Spawn sky lighting objects [...]
#---------------------------------------------------------------

Sun = unreal.EditorLevelLibrary.spawn_actor_from_class(
 unreal.DirectionalLight, location=[0,0,0], rotation=[0,0,0])
Sky = unreal.EditorLevelLibrary.spawn_actor_from_class(
 unreal.SkyLight, location=[0,0,100], rotation=[0,0,0])
Atmosphere = unreal.EditorLevelLibrary.spawn_actor_from_class(
 unreal.SkyAtmosphere, location=[0,0,200], rotation=[0,0,0])
PPV = unreal.EditorLevelLibrary.spawn_actor_from_class(
 unreal.PostProcessVolume, [0,0,400], rotation=[0,0,0])

#---------------------------------------------------------------
# Setting mobility to get correct lighting [...]
#---------------------------------------------------------------

unreal.DataprepOperationsLibrary.set_mobility(
 [Sun,Atmosphere], unreal.ComponentMobility.MOVABLE)
unreal.DataprepOperationsLibrary.set_mobility(
 [Sky], unreal.ComponentMobility.STATIONARY)

#---------------------------------------------------------------
# Find components to set properties [...]
#---------------------------------------------------------------

SunComp = Sun.get_editor_property("light_component")
SunDirComp= Sun.get_editor_property("directional_light_component")
SkyComp = Sky.light_component # R-O
AtmosComp = Atmosphere.sky_atmosphere_component # R-O

PPVSettings = PPV.settings

#---------------------------------------------------------------
# Set properties (examples) [...] whole code in the Notebook
#---------------------------------------------------------------

SunComp.set_intensity(15)
SunDirComp.set_atmosphere_sun_light(True)

SkyComp.set_editor_property("real_time_capture",True)
SkyComp.set_editor_property("cloud_ambient_occlusion",True)

PPV.enabled = True
PPV.unbound = True
PPVSettings.auto_exposure_min_brightness = 0.35
PPVSettings.auto_exposure_max_brightness = 0.6
```

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
