# Peer review of "Designing a Large-Scale Immersive Visit in Architecture, Engineering, and Construction"

_applsci, doi:10.3390/app13053044_

Round 1

Reviewer 1 Report

The manuscript is interesting, providing a comprehensive guide to develop 3D models on a larger scale. However, in terms of academic writing, the manuscript is poor in presenting 

1. the need for such workflow

2. research design is vague, no scientific justification

The author might consider restructuring the manuscript to be 

1. readable in terms of academic English writing

2. provide scientific justification in the software architecture or workflow development- to highlight the contribution of the work. 

Reviewer 2 Report

1.The paper is too long. I feel like reading a technical paper, rather than a journal article. You may shorten the paper by placing the process in diagramatic forms, rather than textual formats.

2. The abstract is fine, but suddenly you have many other purposes of the study in page 8 : Perception? survey? Interview?

3. References should be made when you mention about usages of certain tools, techniques and software, so that readers know where to look for detailed information.

4. You need to improve on the English composition. Exp 1: ( to be bring/) = to be brought). Exp.2 = (external reality? ) + extended reality

5. Arrangement of figures should be consistent with the textual descriptions.

6. I suggest you generate QR codes as embedded figures, to be linked with online videos, so that readers have the chances to experience the immersive content.

7. You may leave detailed of all partners involved to your acknowledgement and real technical report, but for this paper, just focus on the narrative of process you have undergone.

All the best.

Reviewer 3 Report

The issue presented in the contribution is of great interest and topicality and is well-founded. It is presented with a professional approach and knowledge transfer from the university in collaboration with the construction and property development industry.

It is considered that the article could be improved if the authors reordered and renamed the following sections: 1/ "Section 1.3", it is suggested, be replaced by "Section 1.2" and renamed "State of the question". The purpose of this is to locate the gap in the field in which work is being done in order to propose an advance in knowledge or a solution. It is therefore considered that this section should end with the proposal of the "proposed general and specific objectives" in lines 64 and 65. 2/ "Section 1.2" would be replaced by "section 1.3". It is suggested to include a bibliographical reference on which it is based and to rename it as "methodology applied... "3/ Include the process followed to validate the results, if any, interviews with experts, focus groups, etc.

Reviewer 4 Report

- Introduction should be rewritten and restructured as it meant to provide a background to readers on the topic and then narrow it down to research gap and the need for this study

- There is no clear methodology, research design or methods used. 

-  There is no need to write about the software such as Revit and TwinMotion, it is enough to mention that you used them. The paper already too long and reader will lose interest and focus reading unnecessary narratives. 

- Same for the hardware, reduce the narrative, no need to talk about installation advantages and drawbacks 

- highlight the research gaps and the need for this research 

- State clearly the contribution and significance of this study 

- State clearly the research limitation 

- Reduce number of pages by removing unnecessary narratives 
